

# Summer ozone variation in North China based on satellite and site observations

Lihua Zhou[1], Jing Zhang[1], Hui Wang[1], Wenhao Xue[1], Xiaohui Zheng[1], and Siguang Zhu[2]

[1]College of Global Change and Earth System Science, Beijing Normal University, Beijing, China

[2] Collaborative Innovation Center on Forecast and Evaluation of Meteorological Disasters/Key Laboratory of Meteorological Disaster of Ministry of Education, Nanjing University of Information Science and Technology, Nanjing, China

Correspondence to: Jing Zhang (jingzhang@bnu.edu.cn)

**Abstract.** Compared with other regions, air pollution in North China is very serious, especially its levels of fine particulate matter, which are closely associated with the concentrations of polluting gases, such as nitrogen oxides, sulfur oxides, organic

gases, and ozone. Fine particle pollution has been studied in-depth, but there is less known about ozone. This paper focuses on the interannual variability of tropospheric ozone in North China and identifies its influential factors. Our analysis relies on satellite observations (ozone, nitrogen dioxide, sulfur dioxide, carbon monoxide and formaldehyde concentrations) and near-surface data (carbon monoxide, sulfur dioxide, nitrogen dioxide, fine particulate concentrations, temperature, and humidity). Studies have shown that the tropospheric ozone column in North China has been at a high level for the past 3 years, with the

similar time series for temperature and formaldehyde. However, trends in ozone are opposite to those of sulfur dioxide and nitrogen dioxide over this 3-year period. This indicates that the increase in ozone in North China was mainly caused by the increase in temperature and an increase in organic gas content, rather than by nitrogen oxides. Over both temporal and spatial scales, the production rate of ozone appears to be most sensitive to temperature change, as ground observations in Beijing have suggested.

## 1 Introduction

Ozone ($O_3$) in the atmosphere is mostly distributed within the stratosphere, where it absorbs ultraviolet radiation and maintains atmospheric temperature. Consequently, the $O_3$ content in the troposphere is relatively small, being about ten times less. However, tropospheric $O_3$ has more direct and important impacts on human health and ecosystems. In recent years, North China has experienced severe air pollution, especially related to fine particles of pollution gases.

A large range in spatial distribution and long-term temporal changes of $O_3$ are observed in satellite data. Typically, $O_3$ pollution is closely related to other air pollution components, such as $NO_x$ and volatile organic compounds (VOCs) (Sillman et al., 2003), as well as temperature and humidity. Although previous studies have mostly been based on a case study of the $O_3$–VOC–$NO_x$ system sensitivity, there are few large scale studies of range or change (Carrillo-Torres et al., 2017). In this study, the horizontal transport of $O_3$ was not considered. We only considered vertical transport and photochemical reactions because in summer

these are the two major contributors to tropospheric $O_3$ under $NO_x$-limited conditions (Liu et al., 2010). Specific $O_3$ pollution incidents have received much attention and analysis, although long-term studies based on atmospheric compositions obtained from satellite observations are more likely to show a wide range of macro changes. Numerous studies have shown that Ozone Monitoring Instrument (OMI) observations are reliable for assessment of sources, as well as regional and global

characterization of spatiotemporal variability of $O_3$ (Krotkov et al., 2016; Boersma et al., 2009; Boersma et al., 2008). Such data are worthwhile, as there are still some differences between the atmospheric conditions provided by the laboratory and the actual atmosphere. Chemical reactions in the atmosphere are complex, suggesting we need to visualize the relationship between the concentration of $O_3$ and other parameters. Therefore, we introduce temporal and spatial distributions of tropospheric $O_3$ sect. 3.1; sect. 3.2 discusses temporal and spatial distributions of other components; while sect. 3.3 describes relationships

between $O_3$ and other factors based on ground observations.

## 2 Data and methods

Tropospheric $O_3$, sulfur dioxide ($SO_2$), nitrogen dioxide ($NO_2$), and formaldehyde (HCHO) data are from the OMI aboard the EOS Aura spacecraft, launched 15 July 2004. Detailed data descriptions are provided in the OMI Data User's Guide and references therein (Zhou et al., 2017; Huang et al., 2015; Zhu et al., 2014; Gray et al., 2011). The daily retrieved total column

carbon monoxide (CO) is from Measurement of Pollution in the Troposphere (MOPITT), available at https://terra.nasa.gov/about/terra-instruments/mopitt (Heald et al., 2003). The resolution of these data is $360° \times 180°$ (units: mol/cm$^2$). Ground measurements for daily $SO_2$, $NO_2$, 8-h $O_3$ data and other near-ground gas data were obtained by averaging hourly data recorded at an Environmental Protection Station. Such air quality data have been recorded in most cities since 2013.

Daily average humidity and daily maximum temperature data were retrieved from the China Meteorological Administration Meteorological Data Center (http://data.cma.cn/data/ ). Grid data for the Monthly Mean Maximum Temperature at 2 m were obtained for North China from National Centers for Environmental Prediction (NCEP)/National Center for Atmospheric Research (NCAR) Reanalysis. Daily surface solar radiation downwards data were retrieved from the European Centre for Medium-Range Weather Forecasts (ECMWF). Monthly values were derived from the average of the daily value in each grid,

with a spatial resolution $0.75° \times 0.75°$. The study area, located in North China, is shown in Fig. 1. Our statistical information reflects averages of the grid points within the study area. Summer was selected as the study period, because it is the peak time for $O_3$ pollution in North China.

## 3 Results and Discussion





### 3.1 Temporal and spatial distribution of tropospheric ozone

Figure 2 shows the monthly average $O_3$ distributions for the study area; these statistics cover the period from 2005 to 2016. North China is situated within the monsoon area of East Asia. The dominant airflow in winter is a dry cold northwest airstream originating from Siberia, while the dominant airflow in summer is a southeast airstream from warm and humid air masses over

the oceanic/coastal sea areas. As a result, the region's temperature and humidity are closely linked to season. Figure 2 shows that the distribution of $O_3$ is also closely related to season. Throughout the year, $O_3$ levels are lowest in the winter (December/January data), with highest levels in summer (June to August). Spring and autumn are transitional periods. The most ozone-intense period is July. This pattern is consistent with the seasonal distributions of temperature. July $O_3$ levels rose from 2005 to 2016, indicating that $O_3$ levels in North China are continuing to rise annually, following a longer-term trend with

values rising from 44.9 DU in 2006 to 52.6 DU in 2015. This trend is not apparent in winter. Table 1 lists the percentage of tropospheric $O_3$ per month from 2005 to 2016. These data confirm that the lowest $O_3$ levels are in January. For example, lowest $O_3$ levels in January 2006 accounted for only 5.65 % of annual amounts. Similar low values are often recorded in the December to February period, for example, levels accounted for 5.5% and 5.87 % of total amounts in February 2012 and December 2015. In contrast, levels in July 2016 accounted for 11.91 % of the whole year. Because of the increase in $O_3$ pollution incidents in

North China, now occurring during most summers, we focused our study on the summer period. In addition to $O_3$ distributions, we also discuss several other relevant pollutant gases and meteorological factors. Their potential to affect $O_3$ is analyzed to evaluate their various contributions to its spatial and temporal changes in $O_3$ distribution.

### 3.2 Temporal and spatial distributions of other components

3.2.1 Nitrogen dioxide

There are many types of nitrogen oxides ($NO_x$), several of them causing air pollution, especially nitric oxide (NO) and $NO_2$ (Brown et al., 2003; Foy et al., 2015). Except for $NO_2$ , most nitrogen oxides are extremely unstable. Under intense light, moisture or heat, they are converted into $NO_2$ (or NO, which in turn is converted into $NO_2$). Therefore, the nitrogen oxides in the atmosphere are mainly these two species, having a final form of $NO_2$. Therefore, over long periods, the amount of $NO_2$ and the total amount of nitrogen oxides are basically the same. According to previous research, there is a positive correlation

between atmospheric $NO_x$ and $O_3$ concentrations (Scholz and Rabl, 2006). The formation of tropospheric $O_3$ requires a series of complex chemical reactions. NO and $NO_2$ act as catalysts, yielding the final chemical reaction equation: $O + O_2 \rightarrow O_3$, $\frac{[NO_2]}{[NO]} \propto [O_3]$ (Shon et al., 2008). According to satellite observations, $NO_2$ pollution in North China is more serious than in adjacent areas (Gu et al., 2014). The distributions of this pollution and northern industrial areas are well correlated. Annual trends in $NO_2$ are shown in Fig. 3. Over the past 12 years, highest peaks occurred in 2011, reaching 6.97 DU in June, 6.15 DU in July, and 6.14 DU in August. In July, during the period 2005-2011, $NO_2$ concentrations showed a downward trend from



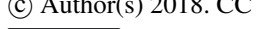

2011 to 2016, as did concentrations in June and August. However, in July, trends in $NO_2$ and $O_3$ are not consistent. Therefore, the increase in $O_3$ over this period is not caused by an increase in nitrogen oxides.

### 3.2.2 Tropospheric sulfur dioxide

$O_3$ and $SO_2$ participate in photochemical reactions (Xie et al., 2005). Oxidation of $SO_2$ by $O_3$ in the presence of water forms sulfate (Ullerstam et al., 2002). $O_3$ has a clear impact on $SO_2$ levels. We conclude that the reduction in $SO_2$ content may reflect an increase in atmospheric oxidants and a reduction in sulfide emissions. Enhanced oxidation properties of the atmosphere, related to increasing $O_3$ content, would increase the sinks of $SO_2$. The annual trend of $SO_2$ in North China is shown in Fig. 4. Quantitative calculations indicate that the overall trend is downward from 2007 to 2011. In July, the average content of $SO_2$ decreased by 43.2 % compared with 2005 values, while June values decreased by 49.1 %, and August values by 43.6 %.

### 3.2.3 Tropospheric carbon monoxide

CO emission from biomass burning was estimated by fire counts (Duncan et al., 2003). The diffusion and migration of CO in the atmosphere is controlled by wind speed (Liu et al., 2003). The solubility of CO in water is very low, and the effect of removing it from the air through wet deposition is not significant. The reaction between CO and oxygen yields: $CO+2O_2 \rightarrow CO_2+O_3$ (Jin, 2008).When the concentration of NO is high, one molecule of CO produces one molecule of $O_3$ (Heald et al., 2003). When the concentration of NO is low, there is no increase in $O_3$; there may even be a loss of $O_3$. In contaminated areas, CO is one of the important precursors to photochemical production of $O_3$. There is a positive correlation between vertical profiles of $O_3$ and CO (Fishman and Seiler, 1983; Choi et al., 2017), where the main fluctuations in concentrations are similar. The positive correlation between CO and $O_3$ suggests that tropospheric $O_3$ is mainly derived from photochemical processes (Tang et al., 2006; Suthawaree et al., 2008; Chin et al., 1994). The spatial distribution of CO is basically like those of $NO_2$ and $SO_2$, all of which have elevated levels in North China. However, interannual trends of CO did not show a clear downward trend. This indicates there is no marked reduction in CO emissions over the study period.

### 3.2.4 Summer ozone concentrations and its relationship to other factors

The order of magnitude of each component in the atmosphere varies greatly and is typically normalized for ease of comparison (Ji et al., 2016). The standardization method used in this study involved normalizing to [0, 1]: $c' = \frac{c_i - c_{min}}{c_{max} - c_{min}}$, where $c_i$ is the input data, $c_{min}$ and $c_{max}$ are the largest and the smallest values in the series (Karanwal et al., 2010; Patel and Shah, 2015). Figure 6 shows temporal trends of various parameters (formaldehyde, temperature and surface solar radiation downwards) in July throughout North China. To compare these parameters, their time series were normalized. Since 2010, the trends in $O_3$ and temperature have been consistent. Meanwhile, HCHO content is positively correlated with VOC content, suggesting these has been consistent changes in all VOC species (Seinfeld and Pandis, 2006). Over the past 12 years, HCHO content has





increased, with a slight decrease following 2012. We conclude that although the concentration of nitrogen oxides decreased over this period, the concentration of $O_3$ did not decrease because VOCs have continued to increase (Duncan et al., 2010). Clearly, temperature and solar radiation are also important factors (Tang et al., 2006).

**3.3 The relationships among ozone concentration and other factors from ground observations**

In cities, there are many components that co-occur with $O_3$. Whether there is chemical correlation between these species is investigated herein. Figure 7 displays the correspondence between $O_3$ content and several other components in Beijing during summer periods of 2013-2016. Clearly, precursor species and meteorological factors affect $O_3$ concentrations, including temperature, humidity, and solar radiation levels. Herein, we have selected two factors to discuss in detail.

Clearly, CO contributes to the formation of $O_3$. But there are other atmospheric components, such as fine particulate matter

$PM_{2.5}$ (Fig. 7a) that may play a role. As the concentration of fine particles has increased, $O_3$ concentration also has rapidly increased. Typically, when the concentration of CO is stable, $O_3$ concentration rises rapidly as $SO_2$ concentrations increase. However, as discussed in sect. 3.2.2, $SO_2$ concentration declines in July, suggesting that any increase in $O_3$ in July is not caused by the decrease in $SO_2$. When the mass ratio of CO and $NO_2$ are close to 1/20, $O_3$ concentrations reach a peak. Meanwhile, when relative humidity reaches 40 %-60 %, CO concentration is greater than 2 μg m$^{-3}$ and $NO_2$ concentration is greater than

40 μg m$^{-3}$, the $O_3$ concentration also peaks. However, the effect of temperature is most pronounced, showing a strong positive correlation with $O_3$ levels (Fig. 7e) (Duncan et al., 2009).

**4 Conclusions**

Based long-term statistics, the tropospheric $O_3$ content of North China is highest in summer (from June to August), accounting for about 33 % of annual amounts, with highest levels in July (Table 1). Tropospheric $O_3$ column concentrations for July are

increasing annually in North China. Near the ground, such periods have been associated with increasing $O_3$ pollution incidents. In contrast, the winter period (from December to February) has the lowest $O_3$ content, accounting for about 18 % of the whole year.

Here, we analyzed temporal and spatial changes of several key atmospheric components to evaluate their impact on $O_3$ formation. Changes in $NO_2$, $SO_2$, and CO were not consistent with changes in $O_3$ levels in July. Acid gas emissions ($NO_2$ and

$SO_2$) decreased annually, while $O_3$ has been increasing annually, suggesting that $NO_2$ and $SO_2$ are not causing $O_3$ pollution to increase. Similarly, there was no obvious increase related to levels of CO. One of the main uncertainties linked to $O_3$ formation is the impact of the emission of organic gases. We considered both CO and HCHO in this study. The impact of reductions of CO was not obvious on $O_3$ levels, which appear to maintain relatively stable levels. HCHO levels observed by satellite show greater fluctuation and are more volatile over this time. However, it is not certain that CO or HCHO are causing the increase

in summer $O_3$ in North China in recent years.





Another parameter to consider is rising temperature. The increase in temperature in July under current climatic conditions is the cause of enhanced pollution related to photochemical reactions. The effect of temperature on the formation of $O_3$ is greater than humidity, although low humidity and very high temperature are not conducive to the formation of $O_3$ pollution. In sunny weather, temperatures closely approximate solar radiation levels, triggering photochemical reactions which produce $O_3$.

However, this process is weakened by rainy or cloudy days. In July 2016, temperature and radiation both declined on average. However, $O_3$ content did not drop. This may be related to levels of fine particulate pollution in the air. We found a strong co-occurrence of $O_3$ and fine particle contamination in this region. Therefore, $O_3$ pollution cannot be separated from hazy production. This also explains why the concentration of $O_3$ is higher than other climatically warm areas in North China in summer, given that the average $PM_{2.5}$ in Beijing in July was 61 μg m$^{-3}$ in 2015, and 69 μg m$^{-3}$ in 2016.

*Author contributions.* Lihua Zhou collected and analyzed the data. Lihua Zhou and Jing Zhang participated in the design of the study and wrote the paper. All authors discussed the research.

*Competing interests.* The authors declare that they have no conflict of interest.

**Acknowledgements.** The work was supported by National Key R&D Program of China (2017YFA0603603), National Science Foundation of China (41575144) and the Fundamental Research Funds for the Central Universities(312231103).

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

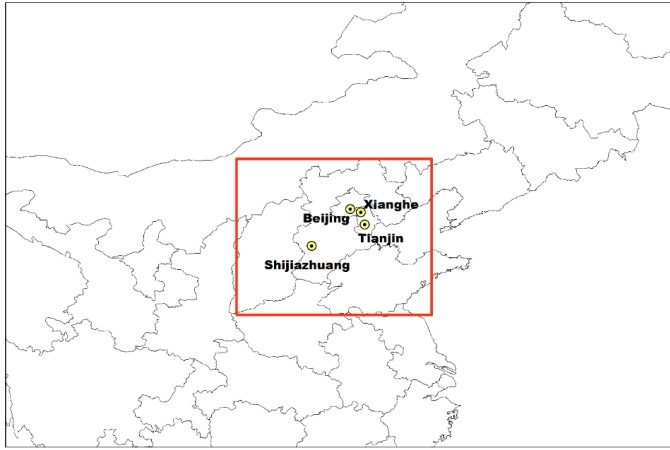

**Figure 1: The focus area of this study is within block, as the North China area referred to in the article.**




**Figure 2: Monthly average ozone vertical column density(VCD)distribution in the troposphere(2005 - 2016)and monthly time series of ozone in North China, as the area in the box in Figure 1 (unit: DU).**





**Figure 3: Average nitrogen dioxide vertical column density distribution in July (2005 - 2016)in the troposphere and summer (from June to August) time series of nitrogen dioxide in North China, as the area in the box in Figure 1 (unit: 10^15 molec cm^-2).**





**Figure 4: Average sulfur dioxide column density distribution in July (2005 - 2016)in the troposphere and summer (from June to August) time series of sulfur dioxide in North China, as the area in the box in Figure 1 (unit: DU).**



**Figure 5: Average carbon monoxide column density distribution in July in the troposphere and summer (from June to August) time series of carbon monoxide (unit: $10^{18}$molec cm$^{-2}$) in North China, as the area in the box in Figure 1.**



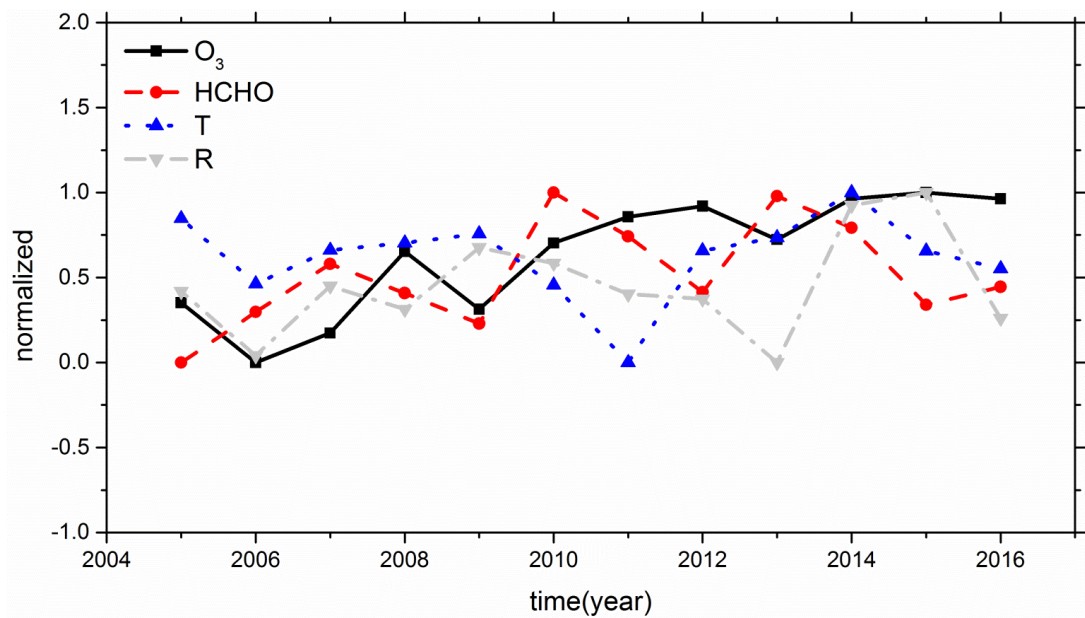

**Figure 6: Correspondence trends standardized of ozone (O₃), formaldehyde (HCHO) and temperature (T), surface solar radiation downwards (R).**



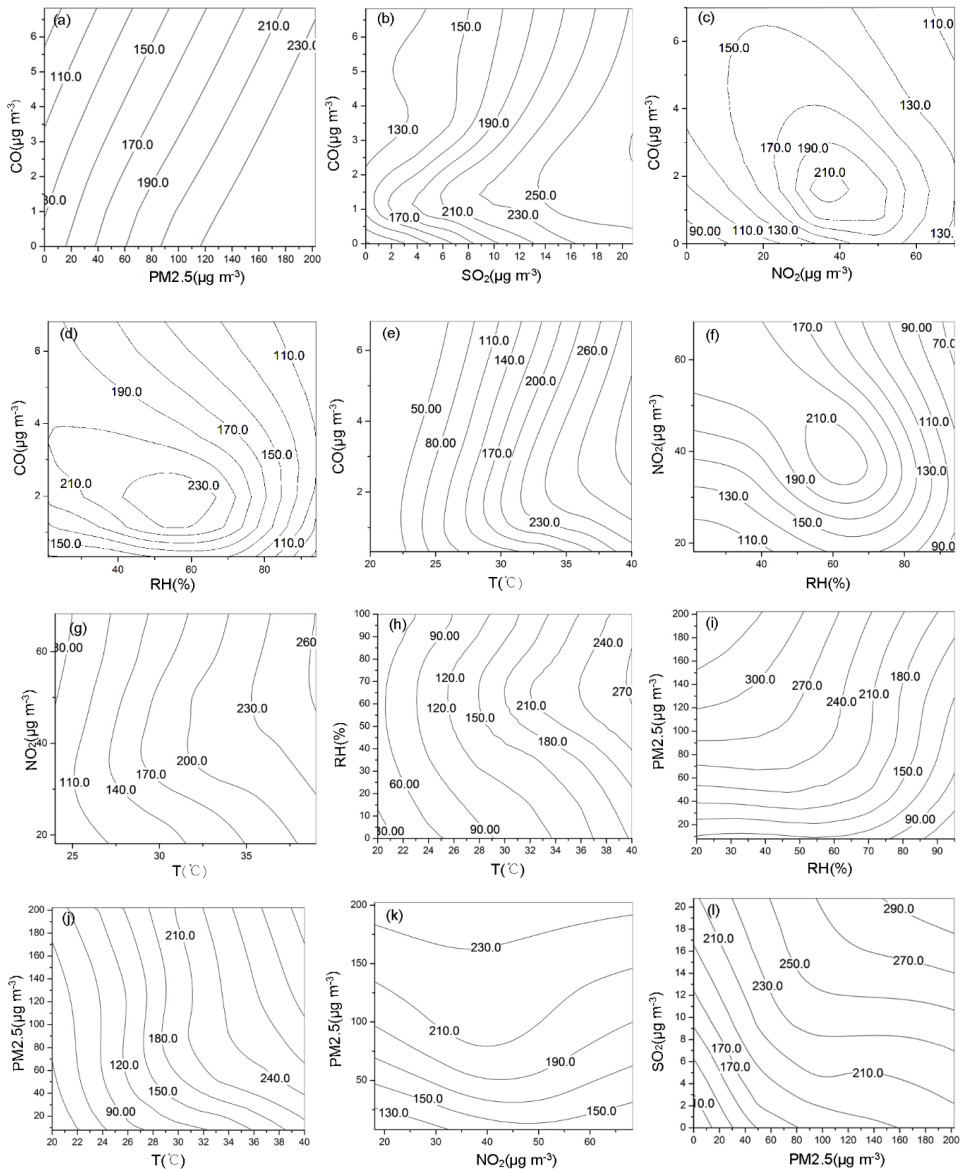

**Figure 7: Correspondence between ozone content (unit: μg m$^{-3}$) and other factors observation in Beijing during summer from 2013-2016.**

**Table 1: Monthly percentage of ozone content and annual content.**

| O$_3$/% | 2005 | 2006 | 2007 | 2008 | 2009 | 2010 | 2011 | 2012 | 2013 | 2014 | 2015 | 2016 |
|---|---|---|---|---|---|---|---|---|---|---|---|---|
| JAN | 6.67 | 5.65 | 5.95 | 6.04 | 6.26 | 5.9 | 6.33 | 6.33 | 5.48 | 6.14 | 6.02 | 6.1 |
| FEB | 6.56 | 6.64 | 5.63 | 6.74 | 6.01 | 6.68 | 6.59 | 5.5 | 6.26 | 6.96 | 6.6 | 6.64 |
| MAR | 7.31 | 6.78 | 6.69 | 7.77 | 7.32 | 7.07 | 7.39 | 7.52 | 6.9 | 7.47 | 7.17 | 7.13 |
| APR | 7.65 | 7.88 | 8.24 | 8.2 | 8.8 | 7.81 | 8.99 | 8.24 | 7.81 | 8.07 | 8.75 | 8.24 |
| MAY | 9.15 | 9.73 | 9.45 | 9.26 | 9.56 | 8.95 | 8.88 | 9.02 | 8.83 | 8.94 | 8.87 | 9.37 |
| JUN | 11.21 | 11.01 | 10.81 | 11.39 | 10.74 | 11.2 | 10.9 | 11.6 | 11.16 | 10.67 | 10.73 | 11.03 |
| JUL | 11.33 | 10.81 | 10.94 | 11.51 | 11.08 | 11.4 | 11.58 | 11.8 | 11.51 | 11.81 | 11.76 | 11.91 |




| | | | | | | | | | | | |
|---|---|---|---|---|---|---|---|---|---|---|---|
| AUG | 10.85 | 11.01 | 11.1 | 10.48 | 10.83 | 11 | 10.76 | 11 | 10.9 | 10.74 | 11.14 | 10.6 |
| SEP | 9.68 | 9.89 | 10.26 | 8.66 | 8.47 | 9.29 | 8.85 | 9.47 | 9.72 | 9.13 | 9.08 | 9.64 |
| OCT | 7.21 | 7.89 | 7.28 | 7.36 | 7.53 | 7.46 | 7.72 | 6.79 | 7.98 | 7.41 | 7.27 | 6.85 |
| NOV | 6.24 | 6.5 | 7.25 | 6.28 | 7.34 | 6.92 | 6.07 | 6.33 | 6.86 | 6.24 | 6.74 | 6.49 |
| DEC | 6.31 | 6.22 | 6.41 | 6.32 | 6.06 | 6.36 | 5.95 | 6.48 | 6.59 | 6.41 | 5.87 | 6 |

**Table 2: $O_3$, $NO_2$, $SO_2$, HCHO, CO based on satellite and temperature (T), surface solar radiation downwards (R) in North China.**

| | $O_3$ (DU) | $NO_2$ ($10^{15}$molec cm$^{-2}$) | $SO_2$ (DU) | HCHO ($10^{15}$molec cm$^{-2}$) | CO ($10^{18}$molec cm$^{-2}$) | T (℃) | solar radiation ($10^6$ J m$^{-2}$) |
|---|---|---|---|---|---|---|---|
| 2005 | 47.57 | 4.29 | 0.64 | 9.52 | 2.62 | 30.68 | 5.96 |
| 2006 | 44.86 | 4.55 | 0.56 | 10.43 | 2.55 | 28.73 | 5.66 |
| 2007 | 46.20 | 5.35 | 0.78 | 11.29 | 2.77 | 29.74 | 5.98 |
| 2008 | 49.89 | 5.02 | 0.59 | 10.77 | 2.64 | 29.95 | 5.88 |
| 2009 | 47.27 | 5.37 | 0.56 | 10.22 | 2.34 | 30.23 | 6.16 |
| 2010 | 50.27 | 5.59 | 0.58 | 12.58 | 2.63 | 28.69 | 6.09 |
| 2011 | 51.45 | 6.15 | 0.74 | 11.79 | 2.56 | 26.37 | 5.95 |
| 2012 | 51.93 | 5.33 | 0.53 | 10.79 | 2.84 | 29.72 | 5.92 |
| 2013 | 50.43 | 4.54 | 0.49 | 12.51 | 2.28 | 30.11 | 5.63 |
| 2014 | 52.27 | 4.68 | 0.44 | 11.94 | 2.53 | 31.46 | 6.36 |
| 2015 | 52.55 | 4.27 | 0.43 | 10.56 | 2.72 | 29.71 | 6.42 |
| 2016 | 52.27 | 3.74 | 0.36 | 10.88 | 2.42 | 29.18 | 5.84 |