# Peer review of "Summer ozone variation in North China based on satellite and site observations"

_Atmospheric Chemistry and Physics, 2018_

## Referee Comment (RC1) · Anonymous Referee #1 · 20 Jul 2018

This manuscript aims to analyze the variation and the influential factors in summertime ozone over North China using multi-satellite and ground-based observations. While topic is of importance to the field, I don't think the authors have presented and interpreted the data in a convincing way. There are a number of issues:

1. I don't think satellite-observed tropospheric ozone can be used as an indicator of surface ozone pollution for several reasons. First, satellite retrieval of tropospheric ozone is very uncertain, as the abundance from stratospheric ozone is so dominated that separating tropospheric ozone from stratospheric ozone is very difficult. There is no discussion on the uncertainties of OMI ozone throughout the paper. Second, tropospheric ozone is not the same as the near-surface ozone. Upper tropospheric ozone is more often considered as a greenhouse gas, while near-surface ozone is considered

as a pollutant. Throughout the manuscript, the authors fail to distinguish tropospheric ozone with near-surface ozone. The authors mention that they use ground-observed ozone, but there are almost no discussions on this. How does the trend in ground-level ozone agree with satellite-observed tropospheric ozone?

2. The trend analysis and the attribution is not convincing to me for a number of reasons. First, the trend analysis is mostly qualitative. The authors suggest an overall increasing trend of ozone from 2005 to 2016, but there is no information on the trend and the statistical significance of the trend. Same is true for other components. Second, the authors try to attribute the trend to other factors by analyzing if the trend in ozone coincides with NO2 or CO or SO2 or HCHO, but I couldn't see how they are correlated just by reading the manuscript. I'd suggest the authors at least provide R2 for the correlation. Even if they are statistically correlated, correlation doesn't mean causality. Third, the authors conclude that VOC, temperature and radiation are the most important factors for increasing ozone, but Figure 6 shows the inter-annual variability of ozone does not follow any of them. I'd suggest the authors provide more quantitative evidence.

3. While the idea of combining satellite and in-situ observations is interesting, I don't see any connection between them. The authors simply analyze them separately. I think ground-based observations could be useful for validating the variation seen from satellite observations.

4. The authors tend to use satellite data without considering the potential issues (e.g. missing values, detection limit) and uncertainties of satellite retrieval. I think the authors should at least discuss how the uncertainties of satellite retrieval would affect the results.

5. The Introduction should be expanded to include the large body of literature behind this topic. For example, how severe is the ozone pollution in China? How have ozone and its precursors changed over the past decades? How have satellite data been used

for studying air pollution (especially ozone) in China?

6. Overall, I think the language of the manuscript should be further polished. There are several grammatical errors, which should be edited carefully.

Specific comments:

Page 1 Line 29: How could you not consider the transport of ozone? Since it's an observation-based study. The spatial patterns you see reflect combining effects of horizontal/vertical transport, chemistry, deposition.

Page 2 Line 5: The references do not use OMI observations to characterize ozone variability. They are focused on NO2, not ozone.

Page 2 Line 15: The description of the satellite data is not clear to me. For example, did you use Level-2 or Level-3 data? Are they daily or monthly products? Which satellite retrieval did you use? I'd suggest the authors refer the relevant papers of the product developers and include more details on the retrieval.

Page 2 Line 16: How could the resolution be 360 x 180 degree?

Table 1: What's the meaning of reporting the percentage of the concentration of Ozone?

Page 3 Line 1: What do you mean by concentrations here? Concentration of ozone? I don't see any correlation between ozone and NO2 in June and August.

Page 5 Line 3: What's the basis of the conclusion?

---

## Referee Comment (RC2) · Anonymous Referee #2 · 27 Jul 2018

This paper investigated summer ozone variation in North China based on satellite and site observations. The conclusion is the production rate of ozone is most sensitive to temperature change, instead of emissions changes. I have concerns about the conclusion of the paper and would like to recommend substantial improvements of the analysis.

General comments:

1. I recommend the authors to polish the manuscript with the help with native speakers. For instance, in the abstract, the 1st sentence is very lengthy; the term of "organic gas content" is not widely used. I guess the authors may want to use NMVOC instead. In introduction, the 2nd sentence is very hard to understand. It is very easy to be lost when reading the manuscript.

[Figure]

2. Introduction. The introduction needs substantial improvements. It is difficult to follow the story line of the introduction. What are the existed findings? What new findings will be expected in this study? How would you like to organize your manuscript?

3. Page 2, line 26. The motivation of selecting summer as the study period is not convincing. I expect seasonal variations in O3 and its precursors. The dominated driver for O3 variations could change over seasons.

4. Section 3.2.4. "We conclude that although the concentration of nitrogen oxides decreased over this period, the concentration of O3 did not decrease because VOCs have continued to increase (Duncan et al., 2010)." It loos risky for me to make the conclusion merely based on the upward trend of HCHO and downward trend of NOx. The authors followed to state that "Clearly, temperature and solar radiation are also important factors (Tang et al., 2006).", without any further details. I'm not sure about what is "clearly" here.

5. The total columns of O3 are used as indicators of surface O3, which looks improper for me.

6. Overall, I feel an in-depth analysis is missing. The manuscript only listed the trends of a few pollutants, which has been documented by existing literatures. The author may want to perform sensitivity analysis using CTMs to validate whether the conclusion is solid.

Specific comments:

1. Page 1, line 28, I don't get the meaning of the sentence.

2. Page 2, line 3, what do you mean by "macro changes"?

3. Page 2, line 13. The grammar seems not proper for "launched 15 July 2004".

4. Page 2, line 16. It looks not right for me to say "The resolution of these data is 360° × 180° ".

5. Page 2, line 17. The data source for daily SO2, NO2 and 8-h O3 data is missing.

6. Page 3, line 23. The statement of "Therefore, over long periods, the amount of NO2 and the total amount of nitrogen oxides are basically the same. " is not correct.

---

## Author Comment (AC1) · 28 Sep 2018

**Responses to the Interactive comment from Referee #1 on "Summer ozone variation in North China based on satellite and site observations" by Lihua Zhou et al.**

**Anonymous Referee #1**

**To reviewer:**

**Thank you very much for your great efforts on our manuscript. We also appreciate the referees for the valuable suggestions and questions.**

This manuscript aims to analyze the variation and the influential factors in summertime ozone over North China using multi-satellite and ground-based observations. While topic is of importance to the field, I don't think the authors have presented and interpreted the data in a convincing way. There are a number of issues:

**1.**

**(1) comments from Referees**

I don't think satellite-observed tropospheric ozone can be used as an indicator of surface ozone pollution for several reasons. First, satellite retrieval of tropospheric ozone is very uncertain, as the abundance from stratospheric ozone is so dominated that separating tropospheric ozone from stratospheric ozone is very difficult. There is no discussion on the uncertainties of OMI ozone throughout the paper. Second, tropospheric ozone is not the same as the near-surface ozone. Upper tropospheric ozone is more often considered as a greenhouse gas, while near-surface ozone is considered as a pollutant. Throughout the manuscript, the authors fail to distinguish tropospheric ozone with near-surface ozone. The authors mention that they use ground-observed ozone, but there are almost no discussions on this. How does the trend in ground-level ozone agree with satellite-observed tropospheric ozone?

**(2) author's response**

Thank you for your comments. Considering the above reasons,We intend to use ground-level ozone observation data to verify tropospheric ozone changes. At the same time, we have added an introduction to satellite tropospheric ozone.

**(3) author's changes in manuscript**

2.1 Satellite data

Tropospheric ozone data is obtained from combined observations of two satellite instruments, Ozone Monitoring Instrument (OMI) and Microwave Limb Sounder (MLS). Daily OMI/MLS tropospheric ozone data were determined by subtracting MLS stratospheric column ozone from OMI total column ozone. Stratospheric column ozone from MLS was spatially interpolated (2D Gaussian/linear latitude-longitude interpolation) each day to fill in between the actual along-track measurements. The monthly means were then determined by averaging all available daily data within each month. The formatting of the data files is 1 degree latitude by 1.25 degree longitude resolution. OMI total column ozone was filtered for near clear-sky conditions by including only measurements when coincident OMI reflectivity was less than 0.3 (https://acd ext.gsfc.nasa.gov/Data_services/cloud_slice/new_data.html).

The correlation coefficient R between OMI/MLS tropospheric $O_3$ and WOUDC ozonesonde tropospheric $O_3$ is 0.92, RMS is 6.0ppbv from April to October. And the deviation is smaller in the location with lower latitude. It indicates closely similar signatures for seasonal cycles and spatial variability from the comparisons of OMI/MLS tropospheric ozone between the climatology and other data products (Ziemke et al., 2006, 2011).

Daily sulfur dioxide ($SO_2$) and nitrogen dioxide ($NO_2$), monthly formaldehyde (HCHO) data are from the OMI aboard the EOS Aura spacecraft, launched on July 15, 2004. The spatial resolution is $0.25°\times0.25°$. Detailed data descriptions are provided in the OMI Data User's Guide and references there in (Zhou et al., 2017; Huang et al., 2015; Zhu et al., 2014; Gray et al., 2011). The daily retrieved total column carbon monoxide (CO) is from Measurement of Pollution in the Troposphere (MOPITT), available at https://terra.nasa.gov/about/terra-instruments/mopitt(Heald et al., 2003). And its resolution is $1° \times 1°$. In our analysis, the missing values are eliminated. Individual satellite observations have large errors, but averaging over long periods of time and large spatial extent can reduce them. And systematic errors affect all data, so the impact on the trend is not obvious.

Fig.2(a) shows the tropospheric ozone from OMI/MLS satellite and the ground observations in North China. The correlation coefficient between them is 0.89. And the statistical results are significant at 0.05. But it seems that the tropospheric ozone peak is one month later than the ground peak. The correlation coefficient between the ground value and the troposphere value in next month is greater, which is 0.93 and significant at 0.05.Thusthere is a high correlation between the tropospheric ozone and the ground ozone concentration.

[Figure]

**Figure 2: The tropospheric ozone vertical column density (VCD), near - surface ozone, monthly temperature at 2 metre (T$_{2m}$ ) and surface net solar radiation (SSR) in North China.**

**(1) comments from Referees**

The trend analysis and the attribution is not convincing to me for a number of reasons. First, the trend analysis is mostly qualitative. The authors suggest an overall increasing trend of ozone from 2005 to 2016, but there is no information on the trend and the statistical significance of the trend. Same is true for other components. Second, the authors try to attribute the trend to other factors by analyzing if the trend in ozone coincides with NO2 or CO or SO2 or HCHO, but I couldn't see how they are correlated just by reading the manuscript. I'd suggest the authors at least **provide R2 for the correlation.** Even if they are statistically correlated, correlation doesn't mean causality. Third, the authors conclude that VOC, temperature and radiation are the most important factors for increasing ozone, but Figure 6 shows the inter-annual variability of ozone does not follow any of them. I'd suggest the authors provide more quantitative evidence.

**(2) author's response**

More data analysis, correlation analysis and significance testing are added. We distinguished interannual and seasonal variation of tropospheric ozone and calculate the correlation coefficient separately.

**(3) author's changes in manuscript**

**3.1 Interannual variation of tropospheric ozone**

Due to lack of data, ground observation data was only obtained after 2014 in North China. Monthly mean tropospheric ozone, temperature at a height of 2 meters, surface solar radiation are available for the period of 2005 to 2016. Fig.2(a) shows the tropospheric ozone from OMI/MLS satellite and the ground observations in North China. The correlation coefficient between them is 0.89. And the statistical results are significant at 0.05. But it seems that the tropospheric ozone peak is one month later than the ground peak. The correlation coefficient between the ground value and the troposphere value in next month is greater, which is 0.93 and significant at 0.05.Thusthere is a high correlation between the tropospheric ozone and the ground ozone concentration.

Previous studies have shown that changes in ozone are the result of a common image of meteorological factors and precursors. Therefore, we first analyze the impacts of two important meteorological factors, temperature and solar radiation on ozone. Fig. 2(b) shows the correspondence between near-surface temperature and tropospheric ozone. The trends for them are very consistent, with a significant statistical correlation coefficient of 0.93. And their annual peaks also appeared at the same time (summer).More details about ozone changes in the summer will be discussed later. The effect of solar radiation on ozone can be seen in Fig. 2(c). The correlation coefficient between the two time series is0.82.Tropospheric ozone peak generally appears 1-2 months later than the solar radiation. But the solar radiation is highly correlated with the ground-level ozone, with a significant statistical correlation coefficient of 0.98.

Surface concentrations of trace gases $NO_2$, $SO_2$, CO are collected for all sites in NCP for the period 2014 -2016.We eliminated the missing values and averaged the data for all sites. The correlation coefficients and significance of the gases with ozone are shown in Table 1. The statistical analysis shows these gases are negatively correlated with the tropospheric and ground-level ozone, and the results are significant (at level of 0.05).This suggests that the ozone pollution and the trace gas pollution might not be concurrent.

The above results are based on monthly averages. To examine the interannual variations of ozone and trace gases, we use the case of Beijing, since more historical data for Beijing is available compared with the surrounding provinces or cities (Hebei, Tianjin, Shandong, Shanxi and Henan).

Figure 3 shows the interannual variations of tropospheric ozone and ground-level trace gases as well as 2 m temperature and solar radiation from 2005 to 2016. Interannual variations in solar radiation and temperature presented nonobvious pattern. There was a rising trend in tropospheric ozone (0.14 DU/a). After 2013, the concentration of CO increased and maintained a stable value. After 2008, the level of $NO_2$ has dropped. Since 2006, $SO_2$ has been declining continuously. Table 2 shows the correlation coefficient between annual tropospheric ozone and ground observations of $NO_2$, $SO_2$, CO, temperature (T), surface solar radiation downwards (R) in Beijing. There is a significant negative correlation between tropospheric ozone and $SO_2$.The same is true for $NO_2$.This indicates that the interannual variation of ozone is opposite to that of $NO_2$ and $SO_2$.The reduction of these two trace gases did not reduce ozone, and even increased the production of ozone. The positive relationship between ozone and CO is weak, and the results are not significant. The annual ozone has a weak negative correlation with annual temperature and solar radiation, and the results are not significant either. Therefore, once the monthly value change is removed, the relationship between ozone and meteorological factors might probably be smoothed and hidden. Therefore, the study of ozone needs to be divided into seasons. And this part is introduced below in Sect. 3.2.

Although temperature and solar radiation are also important factors in photochemical production of $O_3$ (Tang et al., 2006), from the statistical relationship, the positive correlation is weak and not significant between tropospheric $O_3$ and temperature/solar radiation (Table 3). VOCs and NOx are the major ozone precursors. As shown in Table 3, the relationship between $O_3$ and $NO_2$ is not so significant, and the correlation coefficient is 0.05. Another significant factor is formaldehyde with a correlation coefficient of 0.3 with $O_3$. Sulfur dioxide and carbon monoxide are not important ozone precursors.

Studies have shown that the variation trend of tropospheric $O_3$ is rising, with an average growth rate 0.2DU/year. The tropospheric ozone column in North China has been at a high level for the past 3 years. The trend of tropospheric $O_3$ and ground-level $O_3$ is relatively consistent on seasonal changes. If all seasons considered, temperature and solar radiation are the dominant factors affecting ozone. In the summer, there is a significant positive relationship of $O_3$ with satellite observations of HCHO. However, $O_3$ variation trends are opposite to $SO_2$ and $NO_2$ over 2012 - 2016. Since HCHO increases by $0.048 \times 10^{15}$ molec cm$^{-2}$ per year during 2005 to 2016, and $NO_2$ is reduced by $0.90 \times 10^{15}$ molec cm$^{-2}$ per year in summer since 2012. This indicates that the increase in ozone in North China was probably caused by the increase of non-methane volatile organic compounds (NMVOC), rather than by nitrogen oxides. For all seasons, the effects of $SO_2$ and CO on ozone are not significant.

| | | surface $O_3$ | $T_{2m}$ | R | surface CO | surface $NO_2$ | surface $SO_2$ |
|---|---|---|---|---|---|---|---|
| $O_3$ VCD | R* | 8.92E-01 | 9.28E-01 | 8.23E-01 | -8.13E-01 | -8.86E-01 | -8.15E-01 |
| | P | 7.25E-12 | 1.67E-62 | 2.94E-36 | 1.53E-08 | 1.55E-11 | 2.38E-08 |
| surface $O_3$ | R* | | 8.84E-01 | 9.76E-01 | -8.13E-01 | -8.69E-01 | -7.89E-01 |
| | P | | 1.90E-11 | 2.09E-21 | 1.85E-10 | 1.15E-10 | 1.32E-07 |

*$p<0.05$.

**Table 2: Correlation coefficient between annual tropospheric ozone and ground observation $NO_2$, $SO_2$, CO, temperature (T), surface solar radiation downwards (R) in Beijing.**

| | | $SO_2$(mg m$^{-3}$) | $NO_2$(mg m$^{-3}$) | CO(mg m$^{-3}$) | T(K) | R(J m$^{-2}$) |
|---|---|---|---|---|---|---|
| $O_3$(DU) | R* | -0.8747 | -0.7395 | 0.3041 | -0.2458 | -0.3009 |
| | P | 0.0002 | 0.0060 | 0.3366 | 0.2458 | 0.3419 |

*$p<0.05$.

**Table 3. Correlation coefficient value from satellite observations in summer during 2005 – 2016 in NCP.**

| | | $NO_2$ | $SO_2$ | HCHO | CO | Radiation | T |
|---|---|---|---|---|---|---|---|
| $O_3$ VCD | R* | 0.05 | -0.33 | 0.37 | -0.13 | 0.17 | 0.28 |
| | P | 0.76 | 0.05 | 0.02 | 0.44 | 0.33 | 0.10 |

*$p<0.05$.

**(1) comments from Referees**

While the idea of combining satellite and in-situ observations is interesting, I don't see any connection between them. The authors simply analyze them separately. I think ground-based observations could be useful for validating the variation seen from satellite observations.

**(2) author's response**

Ground observations do help to verify satellite observations, so we intend to add this comparison.

**(3) author's changes in manuscript**

Fig.2(a) shows the tropospheric ozone from OMI/MLS satellite and the ground observations in North

China. The correlation coefficient between them is 0.89. And the statistical results are significant at 0.05. But it seems that the tropospheric ozone peak is one month later than the ground peak. The correlation coefficient between the ground value and the troposphere value in next month is greater, which is 0.93 and significant at 0.05. Thus there is a high correlation between the tropospheric ozone and the ground ozone concentration.

[Figure]

**Figure 2: The tropospheric ozone vertical column density (VCD), near - surface ozone, monthly temperature at 2 metre (T$_{2m}$ ) and surface net solar radiation (SSR) in North China.**

**(1) comments from Referees**

The authors tend to use satellite data without considering the potential issues (e.g. missing values, detection limit) and uncertainties of satellite retrieval. I think the authors should at least discuss how the uncertainties of satellite retrieval would affect the results.

**(2) author's response**

The authors appreciate your constructive comments. We will provide uncertainty analysis of satellite data, such as missing values, factors that affect the inversion results. In the calculation process, we eliminate the missing values, which may have a certain impact on the observations.

**(3) author's changes in manuscript**

Tropospheric ozone data is obtained from combined observations of two satellite instruments, Ozone Monitoring Instrument (OMI) and Microwave Limb Sounder (MLS). Daily OMI/MLS tropospheric

ozone data were determined by subtracting MLS stratospheric column ozone from OMI total column ozone. Stratospheric column ozone from MLS was spatially interpolated (2D Gaussian/linear latitude-longitude interpolation) each day to fill in between the actual along-track measurements. The monthly means were then determined by averaging all available daily data within each month. The formatting of the data files is 1 degree latitude by 1.25 degree longitude resolution. OMI total column ozone was filtered for near clear-sky conditions by including only measurements when coincident OMI reflectivity was less than 0.3 (https://acd ext.gsfc.nasa.gov/Data_services/cloud_slice/new_data.html).The correlation coefficient R between OMI/MLS tropospheric $O_3$and WOUDC ozonesonde tropospheric $O_3$is 0.92, RMS is 6.0ppbv from April to October. And the deviation is smaller in the location with lower latitude. It indicates closely similar signatures for seasonal cycles and spatial variability from the comparisons of OMI/MLS tropospheric ozone between the climatology and other data products (Ziemke et al., 2006, 2011).

**(1) comments from Referees**

The Introduction should be expanded to include the large body of literature behind this topic. For example, how severe is the ozone pollution in China? How have ozone and its precursors changed over the past decades? How have satellite data been used for studying air pollution (especially ozone) in China?

**(2) author's response**

We reorganized the introduction and added the corresponding references.

**(3) author's changes in manuscript**

$O_3$ pollution in China is serious. Yangtze River Delta (YRD) is one of the regions experienced serious $O_3$ pollution, with the highest frequency occurring in late spring and early summer (Cheung and Wang, 2001). Pearl River Delta (PRD) is another region with serious ozone pollution (Zhang et al., 2011).North China Plain (NCP) has been not only suffering from severe hazy weather but also one of the regions with serious $O_3$ pollution in summer. It was reported that the high level $O_3$ concentration reached 286 ppbv in the rural region of Beijing (Wang et al., 2006). Most of the research on ozone in NCP was based on model simulations and site observations (Duan et al., 2008;Xie et al., 2008;Shao et al., 2009; An et al., 2012), and lacks long-term sequence presentation. This is the focus of this paper.

Research on long – term changes of ozone pollution is very limited due to the lack of data. In the PRD, the increasing rate of $O_3$was 0.86 ppbv/year from 2006 to 2011(Li et al., 2014). In the NCP, aircraft data indicated boundary-layer ozone with a large increase of 2%/year in the summer time during 1995–2005; the surface daily 1-hour maximum ozone in urban Beijing increased 1.3%/year during 2001–2006 (Tang et al., 2009) and the daily 8-hourmaximum $O_3$ at rural Shangdianzi rose at a rate of 1.1 ppbv/year during 2003–2015 (Maet al., 2016). However, due to the environmental protection regulations in China, the emissions of precursors decreased since 2011 and 2012. For 2010 and 2014, NO-emissions were 1.6 and 1.5 Gg/d in PRD respectively, 3.9 and 3.0 Gg/d in the YRD, and 15.6 and 14.3 Gg/d in NCP. OMI HCHO data shows upward trends in East Asia resulting from anthropogenic effects; however, the trends are negative in the PRD. Areas around the Bohai Sea have become more NO-saturated (Souri et al., 2017).

A large range in spatial distribution and long-term temporal changes of $O_3$ are observed in satellite data. Typically, $O_3$ pollution is closely related to other air pollutants, such as $NO_x$ and volatile organic compounds (VOCs) (Sillman et al., 2003), as well as temperature and humidity. A lot of work has been done on case studies of the $O_3$–VOC–NOx system sensitivity. However, the ozone long term trend is

less noticed and studied (Carrillo-Torres et al., 2017).

In recent years, satellite data have been used to study air pollutants (Safieddine et al., 2016;Jin and Holloway, 2015).Atmospheric environmental satellite loads have nadir and limb scan modes. Limb mode instruments provide vertical column density and vertical profile data. Microwave Limb Sounder (MLS), Tropospheric Emission Spectrometer (TES), Scanning Imaging Absorption Spectrometer for Atmospheric Cartography (SCIAMACHY) are all limb instruments and provide trace gas profiles ($NO_2$, $SO_2$, $O_3$, CO, $H_2O$, NO, HCHO etc.). Ozone Monitoring Instrument (OMI), Measurement of Pollution in the Troposphere (MOPITT) and Total Ozone Monitoring Spectrometer (TOMS) are nadir instruments and provide total vertical column($O_3$, $SO_2$, $NO_2$, HCHO, CO, $CH_4$).These data had be used to study air pollution (Irie et al.,2008), greenhouse gas emissions (Zhang et al.,2013) in China. Satellite data of column density for $SO_2$, $NO_2$ and CO are often used to study air pollution directly. However, due to the particular characteristics of the vertical distribution of ozone (the peak in the stratosphere), it is not appropriate to use the total amount of the nadir column data alone. It is necessary to combine the vertical profile data observed by the limb instrument to study the ozone change in the troposphere.

**(1) comments from Referees**

Overall, I think the language of the manuscript should be further polished. There are several grammatical errors, which should be edited carefully.

**(2) author's response**

Thank for your comments. We have modified and polished the article.

**(3) author's changes in manuscript**

We almost checked the grammar and presentation errors of each sentence in the article. so please read the manuscript.

**Specific comments:**

Thank you for your comments. We really appreciate your precious comments. We have response the specific comments point to point.

**(1) comments from Referees**

Page 1 Line 29: How could you not consider the transport of ozone? Since it's an observation-based study. The spatial patterns you see reflect combining effects of horizontal/vertical transport, chemistry, deposition.

**(2) author's response**

Yes, it is a statement error. Delete it.

**(1) comments from Referees**

Page 2 Line 5: The references do not use OMI observations to characterize ozone variability. They are focused on $NO_2$, not ozone.

**(2) author's response**

The statement does not match the reference

**(3) author's changes in manuscript**

Numerous studies have shown that Ozone Monitoring Instrument (OMI) observations are reliable for assessment of sources, as well as regional and global characterization of spatiotemporal variability of $NO_2$ and $SO_2$ (Krotkov et al., 2016; Boersma et al., 2009; Boersma et al., 2008).

**(1) comments from Referees**

Page 2 Line 15: The description of the satellite data is not clear to me. For example, did you use Level-2 or Level-3 data? Are they daily or monthly products? Which satellite retrieval did you use? I'd suggest the authors refer the relevant papers of the product developers and include more details on the retrieval.

**(2) author's response**

**We will add more detailed data description.**

**(3) author's changes in manuscript**

Tropospheric ozone data is obtained from combined observations of two satellite instruments, Ozone Monitoring Instrument (OMI) and Microwave Limb Sounder (MLS). Daily OMI/MLS tropospheric ozone data were determined by subtracting MLS stratospheric column ozone from OMI total column ozone. Stratospheric column ozone from MLS was spatially interpolated (2D Gaussian/linear latitude-longitude interpolation) each day to fill in between the actual along-track measurements. The monthly means were then determined by averaging all available daily data within each month. The formatting of the data files is 1 degree latitude by 1.25 degree longitude resolution. OMI total column ozone was filtered for near clear-sky conditions by including only measurements when coincident OMI reflectivity was less than 0.3 ([https://acd](https://acd) ext.gsfc.nasa.gov/Data_services/cloud_slice/new_data.html).Thecorrelation coefficient R between OMI/MLS tropospheric $O_3$is 0.92, RMS is 6.0ppbv from April to October. And the deviation is smaller in the location with lower latitude. It indicates closely similar signatures for seasonal cycles and spatial variability from the comparisons of OMI/MLS tropospheric ozone between the climatology and other data products (Ziemke et al., 2006, 2011).

Daily sulfur dioxide ($SO_2$) and nitrogen dioxide ($NO_2$), monthly formaldehyde (HCHO) data are from the OMI aboard the EOS Aura spacecraft, launched on July 15, 2004. The spatial resolution is $0.25°×0.25°$. Detailed data descriptions are provided in the OMI Data User's Guide and references there in (Zhou et al., 2017; Huang et al., 2015; Zhu et al., 2014; Gray et al., 2011). The daily retrieved total column carbon monoxide (CO) is from Measurement of Pollution in the Troposphere (MOPITT), available at https://terra.nasa.gov/about/terra-instruments/mopitt(Heald et al., 2003). And its resolution is $1° × 1°$. In our analysis, the missing values are eliminated. Individual satellite observations have large errors, but averaging over long periods of time and large spatial extent can reduce them. And systematic errors affect all data, so the impact on the trend is not obvious.

**(1) comments from Referees**

Page 2 Line 16: How could the resolution be 360 x 180 degree?

**(2) author's response**

**This is a statement error.**

**(3) author's changes in manuscript**

And its resolution is $1° × 1°$.

**(1) comments from Referees**

Table 1: What's the meaning of reporting the percentage of the concentration of Ozone?

**(2) author's response**

Percentage $=(O_{3(montnly)}/O_{3(total\ year)})*100\%$.

**(3) author's changes in manuscript**

The table 1 was deleted and briefly described.

**(1) comments from Referees**

Page 3 Line 1: What do you mean by concentrations here? Concentration of ozone? I don't see any correlation between ozone and $NO_2$ in June and August.

**(2) author's response**

In summer, the relationship between ozone and nitrogen dioxide has a correlation analysis.

**(3) author's changes in manuscript**

**Table 3. Correlation coefficient value from satellite observations in summer during 2005 – 2016 in NCP.**

|          |      | $NO_2$ | $SO_2$ | HCHO | CO    | Radiation | T    |
|----------|------|--------|--------|------|-------|-----------|------|
| $O_3$ VCD | R*   | 0.05   | -0.33  | 0.37 | -0.13 | 0.17      | 0.28 |
|          | P    | 0.76   | 0.05   | 0.02 | 0.44  | 0.33      | 0.10 |

*$p<0.05$.

VOCs and NOx are the major ozone precursors. As shown in Table 3, the relationship between $O_3$ and $NO_2$ is not so significant, and the correlation coefficient is 0.05. Another significant factor is formaldehyde with a correlation coefficient of 0.3 with $O_3$.

**(1) comments from Referees**

Page 5 Line 3: What's the basis of the conclusion?

**(2) author's response**

According to supplementary data, monthly solar radiation and temperature are positively correlated with ozone concentration. For summer analysis alone, the correlation is not significant.

**(3) author's changes in manuscript**

We first analyze the impacts of two important meteorological factors, temperature and solar radiation on ozone. Fig. 2(b) shows the correspondence between near-surface temperature and tropospheric ozone. The trends for them are very consistent, with a significant statistical correlation coefficient of 0.93.And their annual peaks also appeared at the same time (summer).More details about ozone changes in the summer will be discussed later. The effect of solar radiation on ozone can be seen in Fig. 2(c).The correlation coefficient between the two time series is 0.82.Tropospheric ozone peak generally appears 1-2 months later than the solar radiation. But the solar radiation is highly correlated with the ground-level ozone, with a significant statistical correlation coefficient of 0.98.

Although temperature and solar radiation are also important factors in photochemical production of $O_3$ (Tang et al., 2006), from the statistical relationship, the positive correlation is weak and not significant between tropospheric $O_3$ and temperature/solar radiation (Table 3).

**The reviewer's comments helped us improve the article content greatly, and we thank you again. We express our deep gratitude.**

---

## Author Comment (AC2) · 28 Sep 2018

**Responses to the Interactive comment from Referee #2 on* "Summer ozone variation in North China based on satellite and site observations" by Lihua Zhou et al.**

**Anonymous Referee #2**

**To reviewer:**

**Thank you very much for your great efforts on our manuscript. We also appreciate the referees for the valuable suggestions and questions.**

This paper investigated summer ozone variation in North China based on satellite and site observations. The conclusion is the production rate of ozone is most sensitive to temperature change, instead of emissions changes. I have concerns about the conclusion of the paper and would like to recommend substantial improvements of the analysis.

**General comments:**

**1.**

**(1) comments from Referees**

I recommend the authors to polish the manuscript with the help with native speakers. For instance, in the abstract, the 1st sentence is very lengthy; the term of "organic gas content" is not widely used. I guess the authors may want to use NMVOC instead. In introduction, the 2nd sentence is very hard to understand. It is very easy to be lost when reading the manuscript.

**(2) author's response**

Thank you for your constructive comments. We have followed your advice and polished the article. Every sentence was checked

(3) author's changes in manuscript

In the abstract, the 1st sentence was reorganized as" Air pollution in North China is relatively serious compared with other regions in China. Fine particle pollution has been studied in-depth, but there is less research about long- term troposphere ozone ($O_3$) variation." "organic gas content" is changed as" Non-Methane Volatile Organic Compounds (NMVOC)". In introduction, the 2nd sentence is changed as" Ozone ($O_3$) in the atmosphere is mostly distributed within the stratosphere, and

tropospheric $O_3$ is about one-tenth of stratospheric $O_3$.".

**2.**

**(1) comments from Referees**

Introduction. The introduction needs substantial improvements. It is difficult to follow the story line of the introduction. What are the existed findings? What new findings will be expected in this study? How would you like to organize your manuscript?

**(2) author's response**

Thank for your advice. Upon the reviewer's comments, we rewrote the introduction and improved the background of this research.

**(3) author's changes in manuscript**

Ozone ($O_3$) in the atmosphere is mostly distributed within the stratosphere, and tropospheric $O_3$ is about one-tenth of stratospheric $O_3$. However, tropospheric $O_3$ has direct and detrimental impacts on human health and ecosystems. The production of tropospheric $O_3$ is chemically controlled by the nonlinear relationship of its precursors, volatile organic compounds (VOC) as well as nitrogen dioxide (NOx) (Seinfeld and Pandis, 2012), implying the complexity to fully understand the $O_3$ pollution.

$O_3$ pollution in China is serious. Yangtze River Delta (YRD) is one of the regions experienced serious $O_3$ pollution, with the highest frequency occurring in late spring and early summer (Cheung and Wang, 2001). Pearl River Delta (PRD) is another region with serious ozone pollution (Zhang et al., 2011). North China Plain (NCP) has been not only suffering from severe hazy weather but also one of the regions with serious $O_3$ pollution in summer. It was reported that the high level $O_3$ concentration reached 286 ppbv in the rural region of Beijing (Wang et al., 2006). Most of the research on ozone in NCP was based on model simulations and site observations (Duan et al., 2008;Xie et al., 2008;Shao et al., 2009; An et al., 2012), and lacks long-term sequence presentation. This is the focus of this paper.

Research on long – term changes of ozone pollution is very limited due to the lack of data. In the PRD, the increasing rate of $O_3$was 0.86 ppbv/year from 2006 to 2011(Li et al., 2014). In the NCP, aircraft data indicated boundary-layer ozone with a large increase of 2%/year in the summer time during 1995–2005; the surface daily 1-hour maximum ozone in urban Beijing increased 1.3%/year during 2001–2006 (Tang et al., 2009) and the daily 8-hourmaximum $O_3$ at rural Shangdianzi rose at a rate of 1.1 ppbv/year during 2003–2015 (Maet al., 2016). However, due to the environmental protection

regulations in China, the emissions of precursors decreased since 2011 and 2012. For 2010 and 2014, NO emissions were 1.6 and 1.5 Gg/d in PRD respectively, 3.9 and 3.0 Gg/d in the YRD, and 15.6 and 14.3 Gg/d in NCP. OMI HCHO data shows upward trends in East Asia resulting from anthropogenic effects; however, the trends are negative in the PRD. Areas around the Bohai Sea have become more NO-saturated (Souri et al., 2017).

A large range in spatial distribution and long-term temporal changes of $O_3$ are observed in satellite data. Typically, $O_3$ pollution is closely related to other air pollutants, such as $NO_x$ and volatile organic compounds (VOCs) (Sillman et al., 2003), as well as temperature and humidity. A lot of work has been done on case studies of the $O_3$–VOC–NOx system sensitivity. However, the ozone long term trend is less noticed and studied (Carrillo-Torres et al., 2017).

In recent years, satellite data have been used to study air pollutants (Safieddine et al., 2016;Jin and Holloway, 2015).Atmospheric environmental satellite loads have nadir and limb scan modes. Limb mode instruments provide vertical column density and vertical profile data. Microwave Limb Sounder (MLS), Tropospheric Emission Spectrometer (TES), Scanning Imaging Absorption Spectrometer for Atmospheric Cartography (SCIAMACHY) are all limb instruments and provide trace gas profiles ($NO_2$, $SO_2$, $O_3$, CO, $H_2O$, NO, HCHO etc.). Ozone Monitoring Instrument (OMI), Measurement of Pollution in the Troposphere (MOPITT) and Total Ozone Monitoring Spectrometer(TOMS) are nadir instruments and provide total vertical column($O_3$, $SO_2$, $NO_2$, HCHO, CO, $CH_4$).These data had be used to study air pollution (Irie et al.,2008), greenhouse gas emissions (Zhang et al.,2013) in China. Satellite data of column density for $SO_2$, $NO_2$ and CO are often used to study air pollution directly. Numerous studies have shown that Ozone Monitoring Instrument (OMI) observations are reliable for assessment of sources, as well as regional and global characterization of spatiotemporal variability of $SO_2$ and $NO_2$ (Krotkov et al., 2016; Boersma et al., 2009; Boersma et al., 2008). However, due to the particular characteristics of the vertical distribution of ozone (the peak in the stratosphere), it is not appropriate to use the total amount of the nadir column data alone. It is necessary to combine the vertical profile data observed by the limb instrument to study the ozone change in the troposphere.

In this study, $O_3$ long-term variations are investigated based on atmospheric compositions obtained from satellite observations. Therefore, we introduce tropospheric ozone from OMI/MLS satellite and the ground observations in North China in Sect. 3.1; Sect. 3.2 discusses seasonal variation of tropospheric ozone and temporal and spatial distributions of other components; while Sect. 3.3

describes relationships between $O_3$ and other factors based on ground observations.

**3.**

**(1) comments from Referees**

Page 2, line 26. The motivation of selecting summer as the study period is not convincing. I expect seasonal variations in $O_3$ and its precursors. The dominated driver for $O_3$ variations could change over seasons.

**(2) author's response**

The authors appreciate your comments. Depending on your requirements, we add increase variations in $O_3$ and driver for $O_3$ variations throughout the year. Statistical relationship is as follows.

**Table 1: Correlation coefficient among monthly tropospheric ozone, ground-level ozone, $NO_2$, $SO_2$, CO, T, R in NCP.**

|  |  | surface $O_3$ | $T_{2m}$ | R | surface CO | surface $NO_2$ | surface $SO_2$ |
|---|---|---|---|---|---|---|---|
| $O_3$ VCD | R* | 8.92E-01 | 9.28E-01 | 8.23E-01 | -8.13E-01 | -8.86E-01 | -8.15E-01 |
|  | P | 7.25E-12 | 1.67E-62 | 2.94E-36 | 1.53E-08 | 1.55E-11 | 2.38E-08 |
| surface O3 | R* |  | 8.84E-01 | 9.76E-01 | -8.13E-01 | -8.69E-01 | -7.89E-01 |
|  | P |  | 1.90E-11 | 2.09E-21 | 1.85E-10 | 1.15E-10 | 1.32E-07 |

*$p<0.05$.

**(3) author's changes in manuscript**

Previous studies have shown that changes in ozone are the result of a common image of meteorological factors and precursors. Therefore, we first analyze the impacts of two important meteorological factors, temperature and solar radiation on ozone. Fig. 2(b) shows the correspondence between near-surface temperature and tropospheric ozone. The trends for them are very consistent, with a significant statistical correlation coefficient of 0.93. And their annual peaks also appeared at the same time (summer). More details about ozone changes in the summer will be discussed later. The effect of solar radiation on ozone can be seen in Fig. 2(c).The correlation coefficient between the two time series is 0.82.Tropospheric ozone peak generally appears 1-2 months later than the solar radiation. But the solar radiation is highly correlated with the ground-level ozone, with a significant statistical correlation coefficient of 0.98.

Surface concentrations of trace gases $NO_2$, $SO_2$, CO are collected for all sites in NCP for the period 2014 -2016.We eliminated the missing values and averaged the data for all sites. The correlation coefficients and significance of the gases with ozone are shown in Table 1.The statistical analysis shows these gases are negatively correlated with the tropospheric and ground-level ozone, and the

results are significant (at level of 0.05).This suggests that the ozone pollution and the trace gas pollution might not be concurrent.

**4.**

**(1) comments from Referees**

Section 3.2.4. "We conclude that although the concentration of nitrogen oxides decreased over this period, the concentration of O3 did not decrease because VOCs have continued to increase (Duncan et al., 2010)." It loos risky for me to make the conclusion merely based on the upward trend of HCHO and downward trend of NOx.

The authors followed to state that "Clearly, temperature and solar radiation are also important factors (Tang et al., 2006).", without any further details. I'm not sure about what is "clearly" here.

**(2) author's response**

The authors appreciate you constructive comments. The previous statement may not be rigorous and is a conjecture. The previous conclusions lacked the support of data, so we performed regression statistics and significance tests on the factors involved, and revised and explained the conclusions through tests and calculations.

**Table 3. Correlation coefficient value from satellite observations in summer during 2005 – 2016 in NCP.**

|  |  | $NO_2$ | $SO_2$ | HCHO | CO | Radiation | T |
|---|---|---|---|---|---|---|---|
| $O_3$ VCD | R* | 0.05 | -0.33 | 0.37 | -0.13 | 0.17 | 0.28 |
|  | P | 0.76 | 0.05 | 0.02 | 0.44 | 0.33 | 0.10 |

*$p<0.05$.

**(3) author's changes in manuscript**

If all seasons considered, temperature and solar radiation are the dominant factors affecting ozone. In the summer, there is a significant positive relationship of $O_3$ with satellite observations of HCHO. However, $O_3$ variation trends are opposite to $SO_2$ and $NO_2$ over 2012 - 2016. Since HCHO increases by $0.048\times10^{15}$ molec cm$^{-2}$ per year during 2005 to 2016, and $NO_2$ is reduced by $0.90\times10^{15}$ molec cm$^{-2}$ per year in summer since 2012. This indicates that the increase in ozone in North China was probably caused by the increase of non-methane volatile organic compounds (NMVOC), rather than by nitrogen oxides. For all seasons, the effects of $SO_2$ and CO on ozone are not significant.

Although temperature and solar radiation are also important factors in photochemical production of $O_3$ (Tang et al., 2006), from the statistical relationship, the positive correlation between tropospheric $O_3$ and temperature/solar radiation is weak and not significant (Table 3).

**5.**

**(1) comments from Referees**

The total columns of O3 are used as indicators of surface O3, which looks improper for me.

**(2) author's response**

Thank you so much for your comments. Indeed, the total columns of $O_3$ not equal to surface $O_3$. But their seasonal changes are closely related. For example, they all reach annual peaks in the summer and reach the annual valley in the winter. Further, high concentrations of the total column are often accompanied with high surface $O_3$. The correlation coefficient ($R^2$) between them is reached 0.8, and the results are significant.

**(3) author's changes in manuscript**

Fig.2(a) shows the tropospheric ozone from OMI/MLS satellite and the ground observations in North China. The correlation coefficient between them is 0.89. And the statistical results are significant at 0.05. But it seems that the tropospheric ozone peak is one month later than the ground peak. The correlation coefficient between the ground value and the troposphere value in next month is greater, which is 0.93 and significant at 0.05.Thusthere is a high correlation between the tropospheric ozone and the ground ozone concentration.

**Table 1: Correlation coefficient among monthly tropospheric ozone, ground-level ozone, NO₂, SO₂, CO, T, R in NCP.**

|  |  | surface $O_3$ | $T_{2m}$ | R | surface CO | surface $NO_2$ | surface $SO_2$ |
|---|---|---|---|---|---|---|---|
| O₃ VCD | R* | 8.92E-01 | 9.28E-01 | 8.23E-01 | -8.13E-01 | -8.86E-01 | -8.15E-01 |
|  | P | 7.25E-12 | 1.67E-62 | 2.94E-36 | 1.53E-08 | 1.55E-11 | 2.38E-08 |
| surface O3 | R* |  | 8.84E-01 | 9.76E-01 | -8.13E-01 | -8.69E-01 | -7.89E-01 |
|  | P |  | 1.90E-11 | 2.09E-21 | 1.85E-10 | 1.15E-10 | 1.32E-07 |

*p<0.05.

**6.**

**(1) comments from Referees**

Overall, I feel an in-depth analysis is missing. The manuscript only listed the trends of a few pollutants, which has been documented by existing literatures. The author may want to perform sensitivity

analysis using CTMs to validate whether the conclusion is solid.

**(2) author's response**

Thank you so much for your comments. This article does lack mechanism analysis. The model CTMs is necessary for in-depth analysis. We are going to do this work in the future. This article uses observational data to characterize phenomenon and make simple analysis temporarily.

**Specific comments:**

We appreciate your precious comments. We have modified the paper followed your comments point ot point.

**(1) comments from Referees**

Page 1, line 28, I don't get the meaning of the sentence.

**(2) author's response**

It is changed as "There are a lot of research on a case study of the O3–VOC–NOx system sensitivity. However, the ozone research for long term trend changes are less".

**(3) author's changes in manuscript**

A lot of work has been done on case studies of the $O_3$–VOC–NOx system sensitivity. However, the ozone long term trend is less noticed and studied

**(1) comments from Referees**

Page 2, line 3, what do you mean by "macro changes"?

**(2) author's response**

This change refers to the overall change of the region, not a change in a site.

**(3) author's changes in manuscript**

We rewrote the introduction and removed the sentence.

**(1) comments from Referees**

Page 2, line 13. The grammar seems not proper for "launched 15 July 2004".

**(2) author's response**

This is a syntax error

**(3) author's changes in manuscript**

It is changed as "launched on July 15, 2004".

**(1) comments from Referees**

Page 2, line 16. It looks not right for me to say "The resolution of these data is 360°

×180°".

**(2) author's response**

This is a false representation

**(3) author's changes in manuscript**

It is changed as "1° × 1°".

**(1) comments from Referees**

Page 2, line 17. The data source for daily $SO_2$, $NO_2$ and 8-h $O_3$ data is missing.

**(2) author's response**

From China National Environment Monitoring Centre: http://www.cnemc.cn/

**(3) author's changes in manuscript**

Ground measurements for daily $SO_2$, $NO_2$, 8-h $O_3$ data and other near-ground gas data were obtained by averaging hourly data recorded at China National Environment Monitoring Centre (available at http://www.cnemc.cn/).

**(1) comments from Referees**

Page 3, line 23. The statement of "Therefore, over long periods, the amount of NO2

and the total amount of nitrogen oxides are basically the same. " is not correct.

**(2) author's response**

This sentence is expressed properly, so deleting.

**(3) author's changes in manuscript**

Therefore, the nitrogen oxides in the atmosphere are mainly these two species, having a final form of $NO_2$.

**The reviewer's comments helped us to consider the issue more comprehensively and improved the paper greatly. We express our deep gratitude.**